# Methylene Blue for the Treatment of Radiation-Induced Oral Mucositis during Head and Neck Cancer Treatment: An Uncontrolled Cohort

**DOI:** 10.3390/cancers15153994

**Published:** 2023-08-07

**Authors:** Carlos J. Roldan, David I. Rosenthal, Dhanalakshmi Koyyalagunta, Lei Feng, Keith Warner

**Affiliations:** 1Department of Pain Medicine, The University of Texas MD Anderson Texas Center, Houston, TX 77030, USA; 2McGovern Medical School, The University of Texas Health Science Center at Houston (UTHealth), Houston, TX 77030, USA; 3Department of Radiation Oncology, The University of Texas MD Anderson Cancer Center, Houston, TX 77054, USA; 4Department of Biostatistics, The University of Texas MD Anderson Cancer Center, Houston, TX 77054, USA

**Keywords:** mucositis, head and neck cancer, radiation therapy, methylene blue oral rinse

## Abstract

**Simple Summary:**

Severe pain from oral mucositis in cancer patients affects oral intake, thus disrupting the quality of life. This pathology is thus far not well managed with standard approaches. Although the biggest challenge in clinical practice is pain control, many efforts have unsuccessfully concentrated on prevention. Methylene blue oral rinse is a safe treatment for refractory oral pain in this population. The low cost of MB makes it potentially accessible to patients of all socioeconomic backgrounds.

**Abstract:**

Pain from radiation-therapy-induced oral mucositis during head-neck cancer treatment is aggravated by concurrent chemotherapy and commonly fails traditional treatments. To explore safe and sustainable alternatives, we investigated methylene blue oral rinse to reduce radiation-therapy-related oral mucositis pain. For this, we conducted a retrospective observational cohort study in a tertiary-care academic care cancer center including 85 patients with refractory oral mucositis pain during radiation therapy for head-neck cancer. Changes in pain (scale 0–10), oral function burden (scale 0–6) and requirement for percutaneous endoscopic gastrostomy tube placement were measured. Among 58 patients, 60% received radiation therapy alone and 40% received concurrent chemotherapy-radiation therapy. Methylene blue oral rinse (MBOR) significantly decreased oral mucositis pain for at least 6.2 h (median + SD 8 ± 1.68 before vs. 2 ± 2.20 after; *p* < 0.0001) and oral function burden (3.5 ± 1.33 before vs. 0 ± 0.86 after; *p* < 0.0001). Eleven patients (19%) had percutaneous endoscopic gastrostomy tubes placed before using methylene blue oral rinse; subsequently, four (36%) resumed oral alimentation after methylene blue oral rinse. Two patients (3%) required percutaneous endoscopic gastrostomy tubes despite methylene blue oral rinse. Minimal adverse events were reported (*n* = 9, 15%). Our study showed that methylene blue oral rinse was an effective and safe topical treatment for opioid-refractory oral pain from oral mucositis associated with radiation therapy for head-neck cancer.

## 1. Introduction

Oral mucositis (OM) is a common toxic adverse effect of cancer therapy for the oral, pharyngeal, and esophageal mucosa [1]. The incidence, progression and severity of OM lesions differ between treatment modalities. Patients who undergo chemotherapy have an average incidence of OM near 40%, and the incidence increases with the increasing number of therapy cycles [2]. Patients undergoing radiation therapy (RT) to the head and neck have an incidence of OM of at least 80%, which may reach nearly 100% for those undergoing altered fractionation and/or concurrent chemotherapy regimens [3].

The clinical manifestations of OM include progressive oral pain, dysphagia, and odynophagia. Patients may develop intolerance for oral fluids and caloric intake leading to dehydration, sarcopenia, weight loss, and general debilitation. The burden together with the costs of unplanned admissions, along with the need for enteral hydration and nutrition via gastric feeding tubes, increases costs of care and may interrupt cancer treatment, leading to worse oncologic outcomes [4].

While some strategies to prevent OM have shown promise, they have had limited clinical success and scopes of application [5,6]. Palliation of the pain has been an elusive target and is perhaps the most important goal to achieve. Conventional palliative approaches include topical anesthetics, coating agents, corticosteroids, antihistamine mouthwashes, systemic opioid, non-steroidal anti-inflammatory therapy, and, more recently, gabapentin, whose use has been adopted by various cancer centers despite the lack of evidence attributed to improved pain control in this pathology [7]. The common mainstay of failed conservative therapy for uncontrolled OM pain has been escalating doses of systemic opioids, including intravenous administration and patient-controlled analgesia devices despite safety concerns [8].

Based on its structural composition, topical methylene blue solution has been used to treat a variety of tegument pain syndromes, including anal fissures and genital herpes. When used as a mouth rinse, it has been shown to lower pain scores and lower opioid requirements when compared with patients who received conventional therapy alone for oropharyngeal pain secondary to mucositis because of different cancer therapies [9]. This pain relief has been attributed to different levels of the nociceptive pathways, including peripheral neurolysis, inhibition of nitric oxide synthetase, guanylyl cyclase, and histamine, which, when combined, likely provide antinociception [10].

Our current study endeavored to further substantiate the claim that methylene blue mouthwash is an effective and safe adjunct for the treatment of OM-associated pain in patients with head and neck cancer (HNC) receiving RT with or without concurrent chemotherapy.

## 2. Materials and Methods

### 2.1. Study Setting and Patient Population

This retrospective cohort study was performed at The University of Texas MD Anderson Cancer Center, a tertiary care academic care center, and was approved by the institutional review board with a waiver of consent.

### 2.2. Data Collection

Utilizing billing codes in our electronic medical records, we identified, and retrospectively reviewed study candidates seen in our pain department with OM-associated pain. We selected only those whose symptoms were related to RT for the treatment of HNC. We included patients treated between 1 December 2019 and 1 December 2020.

### 2.3. Selection and Description of Participants

Patients of any age, sex, HNC diagnosis, and stage of disease were included. In addition to RT alone, we included RT combination treatment modalities (e.g., RT combined with chemotherapy or surgery). Completion of follow-up was defined as an adequate medical records documentation from the initial use of methylene blue oral rinse (MBOR) to its discontinuation.

### 2.4. Procedures

Patients with head and neck malignancy undergoing therapy are frequently seen at the pain service for management of OM-related uncontrolled pain. In our practice, those patients are encouraged to continue other modalities of pain management, including systemic analgesics and coating and anesthetic agents in oral rinses, if needed or beneficial. Patients are instructed to swish and gargle 10 milliliters of the MBOR (diluted at 0.05% in water or normal saline solution, a default mix provided by a pharmacy) for five minutes, then to spit, and this is to be done every 6 h until pain control is achieved. Detailed instructions are given about safety, toxicity, side effects and precautions. The MBOR dilution is compounded by either our institutional pharmacy or outside pharmacies, per availability. When MBOR is prescribed, patients are advised to keep a diary of pain levels before and after oral rinsing, medications used, and their effect on their ability to eat and talk.

### 2.5. Outcome Measures

We used validated patient-reported numerical rating scale (NRS) scores for oral pain using a 11-point Likert scale from 0 (no pain) to 10 (worst possible pain). Pain reduction was classified into 3 groups: >5, 2–5, and <2, termed severe, moderate, and mild, respectively. We also documented changes in oral function burden (OFB) based on the modified Harris mucositis-related pain assessment tool measured on a 7-point validated scale from 0 (representing normal) to 6 (representing total inability to eat, swallow, or talk, with each category scored as unable = 2, difficult = 1, and able = 0 and summed). If documented, both the NRS and OFB scores were tabulated before starting MBOR, immediately after the first use, and until pain control was achieved. Information was obtained from the medical records of patients’ encounters at their scheduled follow-up appointments or at their bedside if hospitalized. Notes from patients’ diaries were frequently documented on the medical records.

The incidence, indication and timing of the placement of a percutaneous endoscopic gastrostomy (PEG) feeding tube was also recorded. Other information collected included the location and duration of the pain before MBOR treatment, as well as the duration of the analgesic effect and the time to maximal pain relief. We also reviewed any adverse effects reported and changes in the use of opioids by calculating the morphine-equivalent daily dose (MEDD).

### 2.6. Statistical Analysis

Summary statistics, including the mean, standard deviation, median, and range, are provided for continuous variables such as age, NRS, and OF, and frequency counts and percentages are provided for categorical variables such as gender. The chi-square test was used to evaluate the association between categorical variables. The Wilcoxon signed-rank test was used to evaluate the change in pain scores from before to after treatment. The Wilcoxon rank sum test was used to evaluate the difference in continuous variables. A boxplot was generated as a visual aid to show the changes of continuous variable between patient groups. Statistical software SAS 9.4 (SAS, Cary, NC, USA) and Splus 8.2 (TIBCO Software Inc., Palo Alto, CA, USA) were used for all the analyses.

The dataset, SAS program file, and Splus program files are stored in “Y:\proj\PainMedicine\RoldanCarlos\2019-1143_MethyleneBlueOralRinseForPainCausedByOralMucositisSecondaryToCancerTreatment0819\AnalysisRadiationPts0222”.

## 3. Results

### 3.1. Patient Characteristics

We identified 85 adult patients who were seen in our pain clinic between 1 December 2019 and 1 December 2020 for OM-associated pain related to RT for the treatment of HNC (Figure 1). Among these patients, 27 individuals were excluded; 12 of them were managed by external providers, and others completed cancer treatment and returned home to continue care outside our institution or had incomplete documentation of data related to the inclusion criteria. The final study population consisted of 58 patients. All patients had documentation of oral/throat pain associated with mucositis secondary to RT and were actively receiving oral rinses and concomitant opioid analgesics in oral or transdermal forms. Patients were prescribed MBOR and completed follow-up.

### 3.2. Demographic Characteristics

The demographic information for the 58 included patients is summarized in Table 1. The median age was 56 years, with a range of 30 to 82 years. There were 19 female and 39 male patients. The most common diagnoses were squamous cell carcinoma of the base of the tongue (*n* = 26, 45%) and squamous cell carcinoma of the tonsil (*n* = 14, 24%). The most common combination of cancer therapy used was chemotherapy plus radiation (*n* = 23, 40%), and the least common cancer therapy was RT alone (*n* = 7, 12%), while others had surgery as well (Table 1). The median duration of the OM-related pain before the use of MBOR was 14 days.

### 3.3. Clinical Characteristics

All patients had multi-site National Cancer Institute Common Terminology Criteria for Adverse Events grade 3 mucositis of the oropharyngeal epithelium, and most patients had painful mucosal lesions of the lingual oral area (*n* = 39, 67%) and the oral mucosa (*n* = 28, 48%) (Table 1). Most commonly, patients described their OM-related pain as “burning” (*n* = 50, 86%), and other descriptions of the pain included “ache”, “sore”, “cutting blades”, “scraping”, “needles”, “raw”, and “sharp”.

### 3.4. Efficacy of MBOR

NRS pain scores were reported both before and after MBOR therapy. Most patients had a pain reduction of >5 (*n* = 42, 72%); 11 had a pain reduction of 2–5 (19%), and 5 had a pain reduction of <2 (9%). The mean NRS pain score was 7.59 (standard deviation [SD], ±1.68; median, 8) before MBOR therapy and 2.05 (SD, ±2.20; median, 2) after MBOR therapy. The mean NRS pain score reduction was 5.53 (SD, ±2.44; median, 6; *p* < 0.0001) (Figure 2).

Most patients (*n* = 49, 84%) reported maximum pain relief within the first two doses (within 12 h), and five patients reported maximum pain relief after three–four doses (within 24 h). Four patients claimed a lack of improvement even after several doses of the MBOR. Although most patients reported pain control within the first day, while undergoing RT, most patients had pain recurrence or reported a mean duration of the analgesic effect of MBOR of 6.18 h (SD, ±3.16 h; median, 6 h).

The mean OFB scores (i.e., the ability to talk, chew, and swallow) before MBOR therapy was 3.55 (SD, ±1.33; median, 3.5), and after therapy it was 0.52 (SD, ±0.86; median, 0). The mean OFB score reduction was 3.03 (SD, ±1.54; median, 3; *p* < 0.0001) (Figure 3).

### 3.5. Effect on Opioid Use

The morphine-equivalent daily dose at baseline was documented as a mean of 70.53 (SD, ±76.16; median, 50). Only 22 patients (38%) who reported pain control in the first 24 h of using MBOR continued their systemic opioid regimens at lower doses. Those patients had documented multi-site pain (i.e., at sites other than those affected by OM). Interestingly, the median morphine-equivalent daily dose was 50 for patients younger than 60 years and 30 for patients 60 years old or older (*p* = 0.17). However, because of inadequate documentation, overall conclusions regarding the effect of MBOR on the morphine-equivalent daily dose could not be drawn.

### 3.6. Effect on Percutaneous Feeding Tube Placement

Among the 58 included patients, and for different indications, 11 (19%) had therapeutic PEG feeding tubes placed before using MBOR. Of these patients, 4 (36%) were soon able to ingest nutrition orally after their pain was controlled using MBOR. Only 2 patients (3.44%) had PEG tubes inserted despite receiving MBOR treatment (*n* = 2; 3.44%). One of these patients had a PEG tube placed because of a poor nutritional condition.

### 3.7. Outcomes and Events

A few patients (*n* = 9, 16%) reported adverse events. Three patients experienced an oral burning sensation during their first MBOR treatment. One patient discontinued use of the MBOR because of the pharmacy’s compounding cost (Table 2).

## 4. Discussion

### 4.1. Clinical Efficacy

This study shows that 0.05% MBOR is a safe and effective treatment of OM pain associated with RT or combined chemotherapy and RT for HNC. The mean NRS pain score reduction of 5.53 (SD, ±2.44; median, 6; *p* < 0.0001) after using MBOR reflects its analgesic properties, not achieved with the use of other rinses and systemic analgesics. Whereas NRS pain scores are self-reported, the OFB scores are of more objective clinical relevance (i.e., the ability to talk, chew, and swallow). Therefore, a mean OFB score reduction of 3.03 (SD, ±1.54; median, 3; *p* < 0.0001) objectively supports its clinical efficacy.

Although the clinical indications for therapeutic PEG tubes are multifactorial, not only including odynophagia, the requirement for and dependence on feeding tubes are significant markers of RT or chemo-RT toxicity [11]. Prior studies suggested that as many as 60% to 70% of patients being treated with RT for advanced HNC will require a feeding tube at some point during their course of therapy [12]. In this study, four patients who had PEG tubes previously placed were able to tolerate oral feedings (and stop using PEG) after pain was controlled with MBOR. In addition, only two patients (3%) had PEG tubes inserted despite using MBOR, and one of these patients had the PEG tube placed due to severe malnourishment.

There were no major MBOR toxicities, and compliance in use was high. As the most common toxicity was a transient oral burning sensation, this was tolerated by most who experienced it, and the therapy was continued because the benefit was greater. As this mix is not commercially available, it must be compounded individually, which in some cases led to costs.

### 4.2. Topical Effect of Methylene Blue

Topical methylene blue solutions have been used to treat different tegument pain syndromes, such as anal fissures and pain from genital herpes-simplex infections [13,14]. Multiple publications describe methylene blue as a long-term inhibition of peripheral axons, an antioxidant, and anti-inflammatory [15], hence its analgesic effect [16,17]. When used as an oral rinse, methylene blue is thought to denature free nociceptive nerve endings exposed in the oral lesions of OM, inhibit the nitric oxide inflammatory pathway, and block N-methyl-D-aspartate receptors [18,19,20]. Furthermore, MB may lessen pain transmission by dampening the excitability of neutrons caused by voltage-gated Na-channels [21].

### 4.3. Safety of MBOR

In a recently published trial of MBOR for OM-related pain, patients undergoing RT were excluded from the study because of the unknown ionization effect of MBOR on radiating tissue [22]. However, when topically applied and in such low concentrations, an ionization effect had not been reported and was not a concern to the referring radiation oncologists.

If swallowed, methylene blue is absorbed from the gastrointestinal tract, reaching a peak plasma concentration in less than 2 h [23]. However, this concentration is calculated to be 100-fold less than an equivalent dose administrated intravenously [24]. Therefore, the pharmacokinetics of methylene blue suggest that if it is ingested at diluted concentrations, the plasma levels of methylene blue may be negligible and therefore exempt from pharmacologic interactions or systemic toxicity. Thus, methylene blue is potentially safe to use in the treatment of esophageal mucositis. When swallowed, patients report asymptomatic transient greenish discoloration of the feces and urine. Furthermore, broncho aspiration of MBOR has not been reported, and patients whose risk is elevated based on a speech pathologist evaluation are advised not to use it.

### 4.4. Advantages and Disadvantages of MBOR

Unlike other coating agents containing topical anesthetics, MBOR does not cause local anesthesia, which manifests as intraoral numbness. Thus, methylene blue does not inhibit the gag reflex or alter the taste of ingested meals if patients have a residual taste sensation during RT for HNC. Additionally, methylene blue appears to have an accumulative analgesic effect, as opposed to the short-lasting pain relief provided by other oral rinses.

As the analgesic effect of methylene blue is based on direct contact with the painful sites, a large limitation to its effectiveness is its inability to reach the compromised mucosa of the esophagus and the larynx unless swallowed. In this study, as in previously reported studies, our patients are instructed to swish and spit without swallowing, so posterior and lower pharyngeal and cervical esophageal analgesic effects could not be evaluated.

Although the use of other preventive therapies, such laser and other light therapy, have shown effectiveness, as supported in at least 24 studies [25], the possibility of combining these alternatives with MBOR has not been explored but could further improve the outcome of this population.

### 4.5. Limitations

Inadequacies in data acquisition could not be addressed owing to the retrospective nature of the study. The discrepancy in the size of the comparison groups could not be reconciled. In addition, we could not objectively address patient compliance with the recommendations given.

## 5. Conclusions

Our report demonstrates that MBOR is a safe, low-risk, efficient, and easy-to-use treatment for refractory pain from OM during RT for HNC. In our data, it provided at least 6 h of relief. This therapeutic agent is potentially accessible to patients of all socioeconomic backgrounds in the absence of compounding charges. Larger randomized controlled trials are warranted to evaluate in more detail the pain, functional and nutrition benefits, compliance, reduction in opioid and other systemic analgesic use, reduction in therapeutic PEG requirement, and reduction in costs for unplanned emergency department visits and hospitalizations.

## Figures and Tables

**Figure 1 cancers-15-03994-f001:**
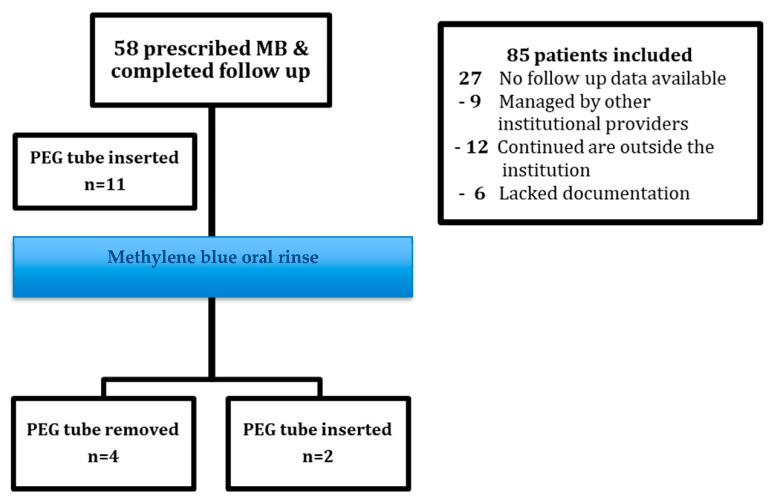
Flowchart of patients included in the study and the effect of MBOR on the outcome of PEG tube placements after its use.

**Figure 2 cancers-15-03994-f002:**
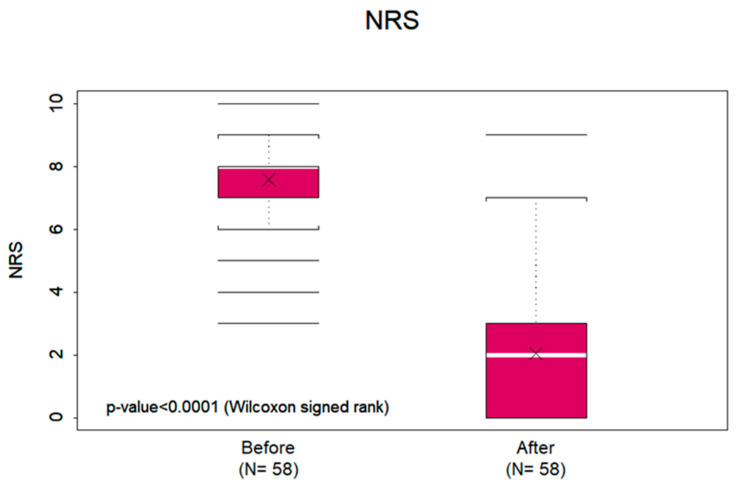
Vertical axis includes numerical rating scale (NRS) pain scores. Horizontal axis describes before and after treatment with methylene blue oral rinse (MBOR). The graph shows the mean (X inside the bar); median (white bar inside the box); interquartile range (entire box); 75th percentile + 1.5 interquartile range or maximum value, whichever is smaller (top bracket at the end of the vertical dotted line); 25th percentile–1 interquartile range or minimum value, whichever is larger (bottom bracket at the end of the vertical dotted line); and outliers (horizontal lines beyond the end of the vertical dotted lines) of the pain scores before (left) and after (right) treatment with methylene blue oral rinse. The pain reduction was significantly different from zero according to the Wilcoxon signed-rank test (*p* < 0.0001).

**Figure 3 cancers-15-03994-f003:**
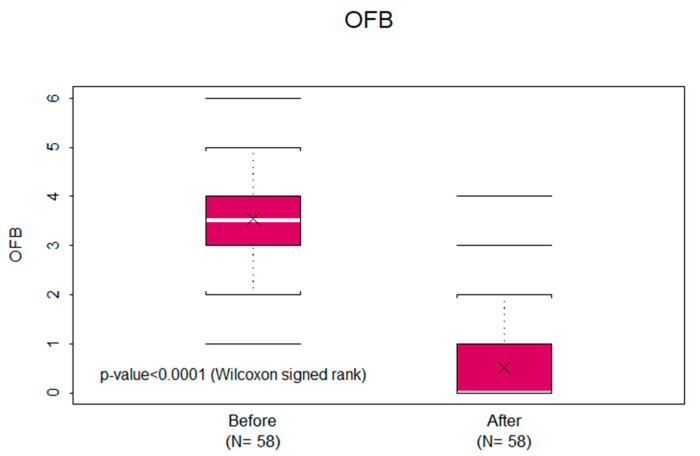
Vertical axis includes oral function burden (OFB) scores. Horizontal axis describes before and after treatment with methylene blue oral rinse (MBOR). The graph shows the mean (X inside the bar); median (white bar inside the box); interquartile range (entire box); 75th percentile + 1.5 interquartile range or maximum value, whichever is smaller (top bracket at the end of the vertical dotted line); 25th percentile–1 interquartile range or minimum value, whichever is larger (bottom bracket at the end of the vertical dotted line); and outliers (horizontal lines beyond the end of the vertical dotted lines) of oral functioning scores before (left) and after (right) treatment with methylene blue oral rinse. The oral functioning score reduction was significantly different from zero according to the Wilcoxon signed-rank test (*p* < 0.0001).

**Table 1 cancers-15-03994-t001:** Patient characteristics (*n* = 58).

Variable	Category	Frequency	Percentage
**Gender**	F	19	33%
	M	39	67%
**Cancer**	SCC of the tongue	26	45%
	SCC of the tonsil	14	24%
	SCC of the oral mucosa	3	5%
	SCC of the nose	2	3%
	SCC of the gums	2	3%
	SCC of the larynx	1	2%
	SCC of the mandible	1	2%
	SCC of the orbit	1	2%
	Adenocarcinoma of the salivary gland	2	3%
	Adenocarcinoma of the sinus	2	3%
	Adenocarcinoma of the tongue	3	5%
	Adenocarcinoma of the palate	1	2%
**Therapy**	Chemotherapy and radiation	23	40%
	Surgery, chemotherapy, and radiation	15	26%
	Surgery and radiation	13	22%
	Radiation alone	7	12%
**Pain location**	Sublingual	39	67%
	Oral mucosa	28	48%
	Soft palate	11	19%
	Oropharynx	11	19%
	Lips	8	14%
	Gums	4	7%

**Table 2 cancers-15-03994-t002:** Reported events.

Events (*n* = 9)	Category	Frequency Count	Percentage
	Inappropriate use	1	2%
	Burning sensation during use	3	5%
	Location of lesion hard to reach with rinse	1	2%
	Discontinued after 1 dose, unknown reason	1	2%
	Discontinued after 2 doses, unknown reason	1	2%
	Discontinued due to cost	1	2%
	Discontinued due to stain and messiness	1	2%

## Data Availability

The data presented in this study are available per request.

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
