# Peer review of "Methylene Blue for the Treatment of Radiation-Induced Oral Mucositis during Head and Neck Cancer Treatment: An Uncontrolled Cohort"

_cancers, 2023, doi:10.3390/cancers15153994_

Round 1
Reviewer 1 Report
This is a good paper on the use of dilute methylene blue mouth washing to relieve pain caused by oral mucositis following radio- and/or chemo-therapy for head and neck cancers. I have the following remarks:
1. The paper is valuable but it is not particularly original (see reference 17).
2. The results on the MB-induced decrease of the pain caused by the oral mucositis, the decrease of the oral function burden (talking, chewing and swallowing), and the reduced necessity of percutaneous feeding tubes are encouraging.
3. The disappointing announcement on page 6 that most patients had pain recurrence, or a (mean) duration of the analgesic effect off ONLY 6.2 hours comes too late in the paper. It should already be announced in the abstract and should figure prominently in the conclusions of the paper. Furthermore this implies that the authors should report on the possibilities of using MB on a much longer time scale. Please note here that in reference 17 (page 18 of 34) it was stated that "But no patients required treatment after 7 days". Please indicate, if you have similar data, i.e. if that was also the case in your patients.
4. It should also be made clear from the start of the paper that the application of MB in these patients does not help for pain that can not be reached by the swishing and gargling of the MB solution, i.e. the oesophagus.
5. An excellent discussion of the analgesic effect of MB and its mechanism is reviewed by S.W. Lee and H.C. Han in Front. Neurosci. 2021, 15:663650. This should probably be included in your list of references. They note the anti-inflammation effect as well as the denervation effect, but they also refer to the SODIUM CURRENT REDUCTION caused by MB. This may be worth while including in your own discussion of the mechanism if you agree with their arguments, i.e. MB may lessen pain transmission by dampening the excitability of neutrons caused by voltage-gated Na-channels.
6. Line 42: Please replace "the incidence increases as does the number of therapy cycles" by "the incidence increases with the increasing number of therapy cycles". That may be slightly clearer.
7. Line 60: What is exactly meant by "Based on the structural composition"?
8. On page 3 line 103 you note (add reference). Apparently you forgot to do this.
9. Figure 1 is not completely clear as to what exactly is happening to the PEG tubes. Please explain in the caption.
10. Page 8, under 4.2, please add the Lee and Han reference cited above, as well as the mention of the ion channel effect.
11. Line 28: "methylene blue oral rinse" should be replaced by "Methylene blue oral rinse (MBOR)". Please check in your paper that whenever you use an acronym for the first time that it is explained in the text!
12. The concentration and purity of your main compound (MBOR) is not defined. Please correct this. Also specify the quantity of liquid being gargled and for how long before being spit out.
13. In the paper by C.J. Roldan et. al. in BMC Medicine 20 (2022) the use different % of MB varying between 0.025% and 0.1%. There was however no significant concentration effect, so that presumably much lower concentrations might be feasible. At those lower concentrations side effects like the burning sensation when gargling might be avoided. Please comment, especially when clearly mentioning the concentration of MB you are using.
14. Please comment more extensively on the use of gabapentin which reduces pain by reducing the excitability of nerve cells in the brain. (Line 57).
15. In Figure 2: On the vertical axis, please add that you are talking about a "Pain Score". On the horizontal axis indicate "Before MBOR" and "After MBOR".
16. In Figure 3: On the vertical axis mention "Oral function burden (OFB)" and on the horizontal axis indicate "Before MBOR" and "After MBOR".
Author Response
July 22, 2023
Cancer journal
Dear Reviewer
We are pleased to submit revisions to our manuscript "Methylene Blue for the Treatment of Radiation-Induced Oral Mucositis During Head and Neck Cancer Treatment: An Un-controlled Cohort”
Thank you for the opportunity to make these changes. I believe we addressed all the changes suggested by your review. Please see our responses to your comments below (highlighted in blue on the manuscript). There were no reviewers’ comments to address
----------------------------------------------------------------------------------------------------------------------
This is a good paper on the use of dilute methylene blue mouth washing to relieve pain caused by oral mucositis following radio- and/or chemo-therapy for head and neck cancers. I have the following remarks:
- The paper is valuable, but it is not particularly original (see reference 17).
Thanks for your comment: The study cited on reference #17 was performed using general information obtained from data generated by the institutional pharmacy. MB was prescribed by different services.
The study was focused only on patients followed by the pain department and receiving radiation therapy. In addition, the insertion of PEG tubes was one of the main variables analyzed since is the turning point for many patients
- The results on the MB-induced decrease of the pain caused by the oral mucositis, the decrease of the oral function burden (talking, chewing and swallowing), and the reduced necessity of percutaneous feeding tubes are encouraging.
Thanks for your comment: The use of MB has changed practice in our institution, it became the first line of treatment for those patients. We keep observing great results.
- The disappointing announcement on page 6 that most patients had pain recurrence, or a (mean) duration of the analgesic effect off ONLY 6.2 hours comes too late in the paper. It should already be announced in the abstract and should figure prominently in the conclusions of the paper.
Comment added on line 28 of the abstract. Also, added on line 293-294 of conclusions section.
Furthermore this implies that the authors should report on the possibilities of using MB on a much longer time scale. Please note here that in reference 17 (page 18 of 34) it was stated that "But no patients required treatment after 7 days". Please indicate, if you have similar data, i.e. if that was also the case in your patients.
Thanks for you inquire: This study showed similar results. Whereas in patients who completed radiation therapy the pain can fully resolve with a few doses of MB, on patients receiving active radiation therapy the still ongoing toxicity of the epithelial mucosa might require continued analgesia until the completion of the treatment.
- It should also be made clear from the start of the paper that the application of MB in these patients does not help for pain that cannot be reached by the swishing and gargling of the MB solution, i.e. the esophagus.
Comment added on line 34 of the abstract. Also highlighted on section 4.4 Advantages and disadvantages of MBOR lines 277-279
- An excellent discussion of the analgesic effect of MB and its mechanism is reviewed by S.W. Lee and H.C. Han in Front. Neurosci. 2021, 15:663650. This should probably be included in your list of references. They note the anti-inflammation effect as well as the denervation effect, but they also refer to the SODIUM CURRENT REDUCTION caused by MB. This may be worth while including in your own discussion of the mechanism if you agree with their arguments, i.e. MB may lessen pain transmission by dampening the excitability of neutrons caused by voltage-gated Na-channels.
Thanks for the suggestion. Reference#15 and #21 were added, edited on lines 252-253 of section 4.2 Topical effect of methylene blue.
- Line 42: Please replace "the incidence increases as does the number of therapy cycles" by "the incidence increases with the increasing number of therapy cycles". That may be slightly clearer.
Thanks for your comment: The editing was done at lines 42-43 of the introduction section
- Line 60: What is exactly meant by "Based on the structural composition"?
Thanks for your comment: That was correct
- On page 3 line 103 you note (add reference). Apparently you forgot to do this.
Thanks for your correction: That error was deleted
- Figure 1 is not completely clear as to what exactly is happening to the PEG tubes. Please explain in the caption.
Thanks for your request: editing was done at lines 146-147 of figure one caption
- Page 8, under 4.2, please add the Lee and Han reference cited above, as well as the mention of the ion channel effect.
Thanks for your comment: The editing was done at section 4.2. Topical effect of methylene blue, lines 252-253, the reference was added
- Line 28: "methylene blue oral rinse" should be replaced by "Methylene blue oral rinse (MBOR)". Please check in your paper that whenever you use an acronym for the first time that it is explained in the text!
Thanks for your comment: Editing was done on line 27 of the abstract
- The concentration and purity of your main compound (MBOR) is not defined. Please correct this. Also specify the quantity of liquid being gargled and for how long before being spit out.
Thanks for your request: editing was done on section 2.4 procedures, lines 95-96
- In the paper by C.J. Roldan et. al. in BMC Medicine 20 (2022) the use different % of MB varying between 0.025% and 0.1%. There was however no significant concentration effect, so that presumably much lower concentrations might be feasible. At those lower concentrations side effects like the burning sensation when gargling might be avoided. Please comment, especially when clearly mentioning the concentration of MB you are using.
Thanks for your request: editing was done on section 2.4 procedures, line 96. This dose has been the recommendation of the compounding pharmacy)
- Please comment more extensively on the use of gabapentin which reduces pain by reducing the excitability of nerve cells in the brain. (Line 57).
Thanks for your request: editing was done at line 57-58 of introduction
- In Figure 2: On the vertical axis, please add that you are talking about a "Pain Score". On the horizontal axis indicate "Before MBOR" and "After MBOR".
Thanks for your request: editing was done at lines 173-174, Figure 2 caption
- In Figure 3: On the vertical axis mention "Oral function burden (OFB)" and on the horizontal axis indicate "Before MBOR" and "After MBOR".
Thanks for your request: editing was done at lines 192-193, Figure 3 caption
The manuscript has been edited following each recommendation provided. Please see attached a version with editions marked in blue
Sincerely
Carlos Roldan MD
Reviewer 2 Report
Oral mucositis, a side effect of tumor therapies, is one of the most important because it significantly worsens the quality of life of patients undergoing therapy and is often the reason for the termination of otherwise promising therapy. Moreover, in the case of the increasingly widespread modern immune checkpoint inhibitor (ICI) tumor therapies, mucosal and skin complications of this kind occur even more often, emphasizing the topic's importance and even guiding the discovery of the mechanism of action. Therefore, any more effective anti-stomatitis therapy that may be promising is an important research direction.
The methylene blue oral rinse (MBOR) therapy proposed and tested in this article can be a straightforward yet effective treatment, as it protects the mucous membrane precisely from the external parts. It is conceivable that an even more beneficial result can be achieved in combination with light therapy. Although, based on this article, it is still quite convincing. A precise therapeutic prescription or application protocol is in great demand on the part of clinicians performing tumor therapy, so the topic can be of great interest in this area. I recommend that in the discussion, the authors deal with the possibility of combining MBOR and light therapy in an additional paragraph based on a literature review. This therapeutic direction would further enhance the reputation of the current article. I recommend the article for publication if the minimum theoretical addition is prepared.
Author Response
July 22, 2023
Cancer journal
Dear Reviewer
We are pleased to submit revisions to our manuscript "Methylene Blue for the Treatment of Radiation-Induced Oral Mucositis During Head and Neck Cancer Treatment: An Un-controlled Cohort”
Thank you for the opportunity to make these changes. Please see our responses to your comments below. Highlighted in yellow in the manuscript.
----------------------------------------------------------------------------------------------------------------------
Oral mucositis, a side effect of tumor therapies, is one of the most important because it significantly worsens the quality of life of patients undergoing therapy and is often the reason for the termination of otherwise promising therapy. Moreover, in the case of the increasingly widespread modern immune checkpoint inhibitor (ICI) tumor therapies, mucosal and skin complications of this kind occur even more often, emphasizing the topic's importance and even guiding the discovery of the mechanism of action. Therefore, any more effective anti-stomatitis therapy that may be promising is an important research direction.
The methylene blue oral rinse (MBOR) therapy proposed and tested in this article can be a straightforward yet effective treatment, as it protects the mucous membrane precisely from the external parts. It is conceivable that an even more beneficial result can be achieved in combination with light therapy. Although, based on this article, it is still quite convincing. A precise therapeutic prescription or application protocol is in great demand on the part of clinicians performing tumor therapy, so the topic can be of great interest in this area. I recommend that in the discussion, the authors deal with the possibility of combining MBOR and light therapy in an additional paragraph based on a literature review. This therapeutic direction would further enhance the reputation of the current article. I recommend the article for publication if the minimum theoretical addition is prepared.
Thanks for the suggestion. Reference #25 was added, edited on of section 4.4 discussion, lines 282-285.
The manuscript has been edited following each recommendation provided. Please see attached a version with editions marked in yellow
Sincerely
Carlos Roldan MD
Round 2
Reviewer 2 Report
Accept in present form.